



**Dynamic Infrared Gas Analysis from Longleaf Pine Fuelbeds Burned in a Wind**
**Tunnel: Observation of Phenol in Pyrolysis and Combustion Phases**
Catherine A. Banach,[1] Olivia N. Williams,[1] Ashley M. Bradley,[1] Russell G. Tonkyn,[1]
Joey Chong,[2] David R. Weise,[2] Tanya L. Myers,[1] Timothy J. Johnson[1]*
[1]Pacific Northwest National Laboratory, Richland, WA USA
[2]USDA Forest Service, Pacific Southwest Research Station, Riverside, CA, USA
*Contact: Timothy J. Johnson, timothy.johnson@pnnl.gov

## 9 0. Abstract

Pyrolysis is the first step in a series of chemical and physical processes that produce flammable
organic gases from wildland fuels that can result in a wildland fire. We report results using a new
time-resolved Fourier transform infrared method that correlates the measured FTIR spectrum to
an infrared thermal image sequence enabling identification and quantification of gases within
different phases of the fire process. The flame from burning fuel beds composed of pine needles
(*Pinus palustris)* and mixtures of sparkleberry, fetterbush and inkberry plants was the natural heat
source for pyrolysis. Extractive gas samples were analyzed and identified in both static and
dynamic modes synchronized to thermal infrared imaging: A total of 29 gases were identified
including small alkanes, alkenes, aldehydes, nitrogen compounds and aromatics, most previously
measured by FTIR in wildland fires. This study presents one of the first identifications of phenol
associated with both pre-combustion and combustion phases, using ca. 1 Hz resolution.
Preliminary results indicate ~2.5x greater phenol emission from sparkleberry and inkberry
compared to fetterbush, with differing temporal profiles.
**Keywords**: Fourier transform infrared, time-resolved, biomass burning, pyrolysis, phenol,
benzene, naphthalene, Pinus palustris, Lyonia lucida, Ilex glabra, Vaccinium arboreum

## 25 1. Introduction

Wildland fire is an important component of many ecosystems and has been used by humans for
several thousand years (Crutzen and Goldammer, 1993; Pyne, 1997; Scott et al., 2014). Many
North American ecosystems have evolved as a result of persistent fire (Barbour and Billings,



2000). The importance of fire in pine forests worldwide including the southern U.S. is well-known
(Agee, 2000; Christensen, 2000). In the U.S., prescribed burning is used on approximately 8
million ha annually to accomplish a variety of forestry and agricultural objectives (Melvin, 2015);
the impact of smoke from these fires has been studied for over 50 years (Chi et al., 1979; Biswell,
1989; Ward and Hardy, 1991; Hardy et al., 2001;). In the southern U.S., forest management
objectives include hazardous fuel reduction, site preparation, improved wildlife habitat, insect and
disease control, enhanced appearance and perpetuation of fire dependent species and natural
communities (Carter and Foster, 2004; Waldrop and Goodrick, 2012). The U.S. Department of
Defense (DoD) uses prescribed burning on approximately 243,000 ha annually for many of these
objectives as well as maintenance of critical training areas (Cohen et al., 2014). Many land
managers rely on fire behavior models to calculate fire movement on the landscape, energy release,
smoke plume development, dispersion and content (Bytnerowicz et al., 2009, Paton-Walsh et al.
2014). However, few fire behavior models account for the plethora of chemical reactions involved
in the fire. The heat transfer processes that take place in the fire environment are also only coarsely
described. In order to improve the use of prescribed burning to accomplish refined objectives,
more detailed description and modeling of the physical and chemical processes in fire are needed
(Cohen et al., 2014).
The chemical phases of wildland fire, described as preheating, flaming, smoldering and glowing
(Ward, 2001) are understood in a chemical sense, but only to varying degrees: While the chemical
effluents of flaming and smoldering phases have been characterized for many ecosystems and fuel
types at different scales (Ward and Radke, 1993), the physics and chemistry of the preheating
(pyrolysis) phase, have fewer studies beyond the bench scale (e.g. Depew et al., 1972;
Dimitrakopoulos, 2001; Susott, 1982; Tihay, 2010). To improve fire application models and to
accomplish the desired fire effects and limit potential fugitive emissions, improved understanding
is thus needed for many fundamental processes, particularly for pyrolysis and ignition in
heterogeneous fuel beds of live and dead fuels that reflect the diversity of vegetation found
worldwide. (Guérette et al. 2018).
Prior to oxidative combustion, biomass thermally decomposes in a heated environment. To study
this decomposition, thermogravimetric analysis has been applied to a small set of plant species
deemed to represent major wildland fuel types (e.g. Burgan and Susott, 1991; Susott, 1982). Others





have determined caloric content of southern fuels which is related to the composition of pyrolysis
products. (Hough, 1969; Behm et al., 2004). However, most such prior work used dried, ground
fuel samples in either an inert or oxidizing environment subject to uniform heating and heat
transfer, (Kibet et al. 2012) thereby eliminating the effects of moisture and heat transfer which are
key fire behavior variables. While pyrolysis and combustion of wildland fuels is known to be a
complex process (Zhou and Mahalingam, 2001), they are often modeled using simple
approximations in the relevant computer codes using the dominant gases of $H_2$, CO, $CO_2$ and $CH_4$.
Heat transfer in a wildland setting is less efficient than in thermogravimetric analysis: The amount
and composition of pyrolyzed species produced depend strongly on heating rate and temperature
and typically consists of oxidized small-molecule gases such as CO, $CO_2$, or $H_2O$, as well as non-
oxidized or partially oxidized species such as $H_2$, $CH_4$, $C_xH_y$, $C_xH_yO_z$ as well as tars. The products
of primary pyrolysis may react in the gas phase at elevated temperatures (i.e., secondary pyrolysis),
which may affect the amount of tar remaining.
This work is part of a larger project to measure and model pyrolysis gases from common wildland
fuels found on DoD installations in the southern United States. (Weise et al. 2018). The project
includes bench-level, laboratory-scale and field plot burns; integrating the results of the field and
laboratory measurements with the modeling results to identify potential improvements that can
enhance our understanding of pyrolysis and ignition in wildland fuels. During the course of the
project Fourier transform infrared (FTIR) technology has been used on several occasions to non-
intrusively measure the composition and concentration of the pyrolysis gases. This includes
identifying the gases liberated by: i) heating single leaf samples from several common southern
fuels using different heating modes in a pyrolyzer and in a simple flat-flame burner system, (Amini
et al., 2019; Safdari et al., 2020) ii) heating shrubs in prescribed burns at Ft. Jackson, South
Carolina, (Scharko et al., 2019a, b) and iii) heating nursery plants with flames from longleaf pine
needle fuel beds inside a wind tunnel (Aminfar et al., 2019). In order to achieve the goal that the
results be applicable to prescribed burns, a key focus has been linking the bench scale, wind tunnel
and field data to the models using realistic values and identities for the pyrolysis gases. Chemical
analysis of the foliage and results of the bench scale tests so far suggest that describing wood
pyrolysis may not be suitable for foliage fuels (live and dead) (Jolly et al., 2012; Jolly et al., 2016;
Matt et al., 2020); to date pyrolysis and ignition of wildland fuels have typically been based on
results for only cellulose or wood (e.g. Varhegyi et al., 1994; Di Blasi, 2008). We have extended





the number of southern fuels examined under more realistic conditions (Amini et al., 2019; Safdari
et al., 2018) including bench-scale measurements of burning individual leaves of plants reported
in this paper. A subset of the bench-scale plant species has been burned in a small wind tunnel to
bridge from the bench-scale to pyrolysis measurements made in the field. The wind tunnel
measurements were set to emulate the larger scale experiments with FTIR instruments and canister
samples in 0.1 ha prescribed burns at Ft. Jackson in May 2018.
Flames in laboratory and field experiments tend to be turbulent in nature meaning there is
inevitably some cross-contamination of the pyrolysis gases with flame gases produced in the
combustion reactions. It is necessary to decouple the phases so as to better understand the discrete
pyrolysis and combustion processes.  The same can also be said of the many studies conducted at
larger facilities such as the Fire Sciences Laboratory (FSL) in Montana, USA.  The FSL has long
path length optical cell coupled to an FTIR (Yokelson et al., 1996, Burling et al., 2010), as well as
many other powerful analytical methods such as proton-transfer mass spectrometry (Christian et
al., 2004; Warneke et al., 2011; Yokelson et al., 2013).  The FTIR and many other systems at the
Missoula FSL have made first detections for dozens of chemical species and pioneered the science
of biomass burning in many regards.  But because the sampling platform is 4 m above the floor,
there is mixing of gases from different phases. The combustion and smoldering phases are typically
easier to differentiate, primarily via the intrinsic diagnostic of the modified combustion efficiency
(MCE, Ward and Hao, 1991) which is a measure that is not independent of the composition of
smoke (Weise et al 2020). The same ambiguity as to the nature of the phase of fire also applies to
extractive methods whereby a sampling device attempts to capture pre-combustion phase gases.
Such sampling systems, typically connected to a field canister are effective but are subject to
vagaries of sniffer gas inlet placement, i.e. proximity to the pyrolyzing plant. (Scharko et al.
2019a,b).
In this paper we describe use of an FTIR with a dynamic probe to temporally isolate, identify and
quantify some of the early-stage/pyrolysis gases from burns at a mid-scale laboratory facility.
During three measurement episodes, experiments were conducted at the Riverside Fire Lab (RFL)
in a wind tunnel using fuel beds composed of longleaf pine needles and the live plants inkberry
(*Ilex glabra* (L.) A. Gray), fetterbush (*Lyonia lucida* (Lam.) K. Koch), sparkleberry (*Vaccinium*
*arboreum* L.) and blueberry (*V. darrowii* Camp).  Multiple methods were used such as quantum-



cascade lasers (Phillips et al., 2020), gas chromatography-mass spectrometry as well as FTIR with
the overall objectives of: i) using careful chemometric extraction from the acquired data to see
what pyrolysis species can be identified by the techniques; ii) using the various methods to
determine the degree of oxidation or combustion, i.e. pyrolysis characterization; iii) making first
attempts to quantify the rates of evolution of pyrolysis products for certain plant species; and
ideally; iv) determining if differences exist between the pyrolysis emissions / temporal profiles for
different plant species. We take advantage of the time-resolved capabilities, as well as high
resolution specificity offered by IR spectroscopy and couple these to the flame/solid fuel
temperature diagnostics of an IR camera to analyze the emissions from a subset of 21 RFL burns.

## 2. Experimental

*2.1 Wind Tunnel and Experimental Configuration*

During November 2017, February 2018 and November 2018 a total of 88 laboratory scale burns
were conducted at the USDA Forest Service Pacific Southwest Research Station in Riverside,
California. The laboratory includes a wind tunnel ca. 3 m long and 1 m wide which was set up to
simulate a forest floor of litter and live plants. Fuel beds composed of 1 kg of dry longleaf pine
needles and various combinations of fetterbush, sparkleberry, blueberry and inkberry (Fig. 1a)
were burned under either "no wind" or 1 m s$^{-1}$ conditions. Fuel moisture content and mass loading,
ambient temperature and relative humidity in the tunnel were varied between experiments; fuel
beds were ignited with a line fire which propagated the length of the fuel bed as seen in Figure 1b.
Multiple analytical techniques were used to study the fire characteristics as well as the gas
effluents: thermocouples, Schmidt-Boelter flux sensor, nadir thermal IR camera and background-
oriented Schlieren photography (Aminfar et al 2019) to estimate heat transfer / air flow around the
plants, canister samples analyzed by GC/FID, quantum cascade (QC) infrared laser spectroscopy,
(Phillips et al., 2020) as well as broadband Fourier transform infrared (FTIR) spectroscopy. A
schematic overview of the experimental setup is seen in Figure 1c.

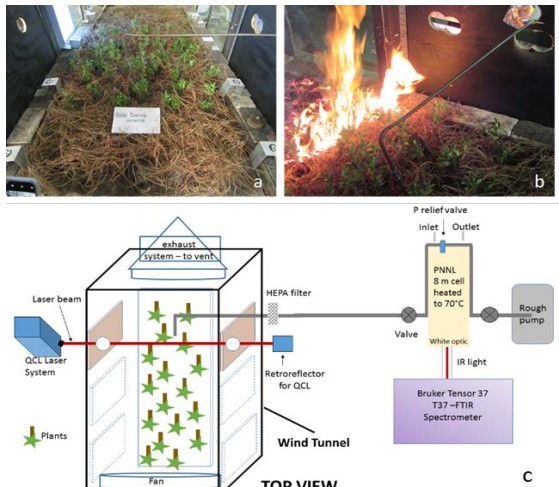

**Figure 1.** a) Overhead view of wind tunnel down its length with longleaf pine needles and interspersed inkberry plants; b) flame front progressing down the wind tunnel with FTIR extraction tube visible; c) cartoon (top view) of experimental layout with laser and FTIR systems.

Gas samples were pumped into the cell / FTIR instrument via a stainless steel probe mounted above the plants (Fig. 1b). Gas from additional probes was pumped into canisters for offline analysis using gas chromatography. Sixty-six or seventy-four live plants were distributed within the longleaf pine needles in ceramic holders. Figure 1a shows the configuration of the fuel bed with instrumentation for *in situ* analysis. Plant species were prepared on site and samples of dry and live fuel were clipped to determine fuel moisture content prior to each burn set. The experiments were set under varying fuel bed and environmental conditions. See Table 1 for the conditions in the 21 experiments presented in this paper.

**Table 1.** Summary of burn schedule for November 2018 studies including burn number, date and time, fuel description, acquisition method and resolution used for wind tunnel experiments under 1 m s$^{-1}$ imposed air flow. Geometric mean flame spread rate = 0.01 m s$^{-1}$. The FTIR acquisition methods are described in the text.





| Burn number | Date (2018) | Local ignition time (PDT) | Local finish time (PDT) | Plant species | Acquistion method | Resolution ($cm^{-1}$) |
|---|---|---|---|---|---|---|
| 76 | 30-Oct | 11:48:01 | 11:52:00 | inkberry | static | 0.6 |
| 77 | 30-Oct | 14:19:10 | 14:23:37 | fetterbush | dynamic | 2.0 |
| 78 | 30-Oct | 15:12:30 | 15:16:33 | sparkleberry | static | 0.6 |
| 79 | 30-Oct | 16:17:00 | 16:21:10 | inkberry | dynamic | 2.0 |
| 80 | 31-Oct | 9:32:00 | 9:35:45 | sparkleberry | static | 0.6 |
| 81 | 31-Oct | 10:35:00 | 10:38:52 | fetterbush | dynamic | 1.0 |
| 82 | 31-Oct | 11:30:30 | 11:35:15 | sparkleberry | static | 0.6 |
| 83 | 31-Oct | 13:19:00 | 13:22:58 | inkberry | dynamic | 1.0 |
| 84 | 31-Oct | 14:12:15 | 14:16:30 | fetterbush | static | 0.6 |
| 85 | 31-Oct | 15:30:30 | 15:34:24 | fetterbush | dynamic | 2.0 |
| 86 | 1-Nov | 9:30:00 | 9:33:02 | sparkleberry | dynamic | 1.0 |
| 87 | 1-Nov | 10:40:00 | 10:42:49 | inkberry | dynamic | 1.0 |
| 88 | 1-Nov | 11:40:00 | 11:42:59 | fetterbush | static | 0.6 |
| 89 | 1-Nov | 13:35:00 | 13:38:48 | inkberry | static | 0.6 |
| 90 | 1-Nov | 14:45:00 | 14:49:47 | sparkleberry | static | 0.6 |
| 92 | 2-Nov | 9:30:00 | 9:34:05 | inkberry | dynamic | 0.6 |
| 93 | 2-Nov | 10:41:15 | 10:45:44 | fetterbush | dynamic | 1.0 |
| 94 | 2-Nov | 11:28:15 | 11:32:28 | sparkleberry | static | 0.6 |
| 95 | 2-Nov | 13:42:45 | 13:46:17 | sparkleberry | static | 0.6 |
| 97 | 2-Nov | 15:38:38 | 15:41:40 | sparkleberry | dynamic | 0.6 |





## 2.2 Instrumentation

Gases were extracted from the burns via 3/8" stainless steel tubing, HEPA filtered to eliminate tar
and char contamination and pumped into an 8-meter White cell (Bruker A136, 2.2 liter volume)[1]
housed inside a Bruker Tensor 37 spectrometer (Figure 1c). The extractive probe was placed
directly above a plant as close as possible to the foliage. To prevent analyte/tar condensation, both
transfer tubing and the gas cell were heated to ~50 °C using heating tape/voltage regulator and a
cell heating shroud, respectively. A thermocouple was suspended into the White cell to record the
gas temperature for subsequent spectral analysis, with pressure gauge mounted atop the cell. Prior
to data collection, the White cell was aligned using the FTIR's Ge/CaF$_2$ beamsplitter and W-lamp
source. Once aligned, these were replaced with a Ge/KBr beamsplitter and mid-IR globar source,
along with a mercury cadmium telluride detector, configuring the Tensor 37 to record spectral data
from 7500 to 500 cm$^{-1}$.
The FTIR system was tested for leaks, followed by a gas cell path length calibration using purified
isopropyl alcohol (IPA - Sigma Aldrich 99.5%). Ten spectra with IPA pressures between 0.6 and
10.5 Torr were recorded to 0.1 Torr using an MKS KF15 pressure transducer. The integrated area
of the 3515-3290 cm$^{-1}$ spectral domain (Bruker OPUS 5.5 software) along with recorded
temperatures and pressures were used to create a Beer-Lambert plot (Scharko et al. 2019a). Using
the integrals from the ten recorded spectra the cell path length was determined to be 6.5 ± 0.2 m.
Wavelength calibration of the infrared data was achieved after the fact using a series of 30 water
rotational-vibrational lines from the PNNL gas-phase database. (Sharpe et al. 2002; Williams et
al., 2013). FTIR interferograms were acquired using double-sided, forward-backward acquisition;
these were apodized using a Blackman-Harris 3-Term function and phase corrected with Mertz's
method prior to Fourier transformation.

## 2.3 Infrared spectral acquisition

Two data acquisition modes were used to analyze the burn gases: an extractive (or static) mode
and a dynamic mode. In the extractive mode the gas flowing through the White cell was isolated;
the inlet/outlet valves were simultaneously closed such that the emitted gases were isolated in the

---

[1] The use of trade or firm names in this publication is for reader information and does not imply endorsement by the U.S. Department of Agriculture of any product or service.



cell at a desired pressure (ca. 740-700 Torr for high pressures, and 430-400 Torr for lower pressure
measurements). The valves were closed just prior to the flame front reaching the probe, attempting
to capture pre-combustion phases including evaporation and pyrolysis. The goal of the extractive
mode was to obtain a higher fidelity "snapshot" for a given point in time of the burn; more data
were averaged longer at higher spectral resolution allowing for detection of more gaseous species
with higher sensitivity. (Scharko et al., 2019a). The dynamic mode measurements had fewer scans
at lower resolution to capture changing chemical identities/composition corresponding to different
fire phases (pyrolysis, flaming combustion, smoldering combustion), achieving temporal
resolutions of the order of ca. 1 Hz.
Of the 21 burns, 10 were recorded using the static method. The static experiment spectra were
recorded using the full resolution of the spectrometer ($0.6$ cm$^{-1}$), a 2 mm Jacquinot stop and double
sided, forward-backward acquisition. Due to the higher resolution and lower light throughput,
acquisition time was extended by averaging multiple scans for a full 30 minutes, resulting in vastly
improved signal/noise ratios (SNR). For analysis of such complicated gas-phase mixtures, infrared
spectral resolutions of $1.0$ cm$^{-1}$ or better have been demonstrated to be advantageous (Burling et
al., 2010, Akagi et al., 2014, Scharko et al., 2019a). While one goal was to isolate gases to include
only the pyrolysis and pre-combustion phases, one vagary of the technique involved the timed
closing of the valves relative to arrival of the flame front approaching the inlet. If the valves were
shut too early, the captured emissions would consist of only (warmed) ambient gas before onset
of thermal degradation of the solid fuel, as opposed to the desired pyrolysis phase. Conversely, if
shut too late, flaming, or possibly even smoldering conditions would be sampled.
The second method was the dynamic mode whereby the OPUS software was used to continuously
collect interferograms throughout the duration of the burn, capturing the chemical compositions
associated with different phases, e.g. volatilization, heating, pyrolysis, flaming or smoldering
combustion. Fourier transformation of the interferograms occurred after the burns to yield faster
acquisition times. The dynamic acquisition mode was used in combination with thermal IR video
imaging recorded from above the flame bed to help synchronize spectral acquisition to the various
burn phases for a total of 11 burns. Instead of averaging for 30 minutes, the dynamic method
allowed for 40-80 continuous interferometer scans (differing on the duration of the burn), and
yielded a spectrum every 1.5 s for data taken at $1.0$ cm$^{-1}$ resolution, every 0.79 s for data at 2.0 cm$^{-}$
$^{1}$ resolution, and every 2.5 s for $0.6$ cm$^{-1}$ resolution spectra. Data acquisition began as the flame





front encroached upon the extractive probe and continued until the flame had passed. Due to the
faster acquisition rate these spectra are significantly noisier than the data collected using the
extractive method. To compare results from the static and dynamic modes, fires 87 and 89 will be
presented. The 2 m length fuel beds for both experiments 87 and 89 consisted of 1 kg longleaf pine
needles with interspersed inkberry plants.
For time synchronization it was necessary to quantify the time lag from the time the emissions
enter the extractive probe to midpoint in their flow through the White cell. A flow rate test was
thus conducted using freon gas, CFC-11 (trichlorofluoromethane) which is comparable in
molecular weight to the heavier gases detected by the FTIR. Figure 2 shows such a test of CFC-
11 being introduced with spectra recorded every 0.79 s using 2.0 cm$^{-1}$ resolution. The time from
introduction of the freon at the extractive probe (t=0, scan 0) to first appearance in scan 4 (maroon
trace) was 3.2 seconds. The freon spectra had maximized at scan 6 (red trace) for a total Δ t = 4.8
s lag from the probe to the instrument. With this information, FTIR time stamped data were then
adjusted to reflect the 4.8 second time delay which was used when correlating the spectral data to
the visual and thermal IR video images.

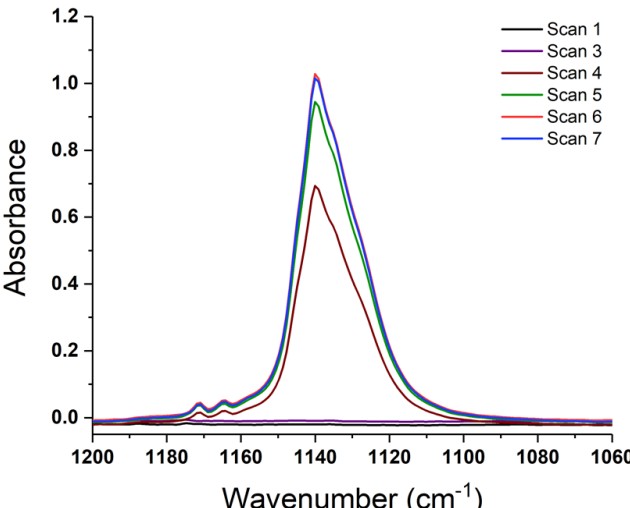


**Figure 2:** Dynamic spectra recording introduction of CFC-11 from extractive probe to gas cell. Scan 0 (not
shown) represents start of spectral acquisition/freon release near probe. Spectra produced every 0.79 s. First
observation of freon occurs with scan 4; maximum absorbance of CFC-11 and stabilization occurs at scan 6.
*2.4 Spectral Analysis*
A combination of software was used for the post-acquisition spectral analysis and confirmation of
the species observed during the campaign. The MALT5 software (Griffith, 2016) utilizing both



HITRAN line-by-line data (Gordon et al., 2017) as well as the PNNL 50 °C gas-phase reference
spectra (Kochanov, 2019; Johnson et al., 2006, 2010) as input libraries was used to identify and
quantify vapor-phase chemicals in the spectra.  Spectra were compiled into parameter files and
analyzed by the MALT software using parameters including pressure, temperature, pathlength,
resolution, as well as estimated initial values for chemical mixing ratios. The software generates a
spectrum to simulate the measured spectrum, adjusting mixing ratios until the residual between
the simulated and measured spectra is minimized.  To confirm the species were actually present,
each spectrum generated by MALT was input to OPUS and subtracted from the measured
spectrum; the target compound was purposefully omitted from the subtraction process to visually
inspect if the omitted compound was in fact present (see e.g. Figure 5).

### 253    3.   Results and Discussion

*3.1 Analysis of Static Spectra*
Ten spectra were recorded from different burns using the static mode with the gas cell valves
closed simultaneously; gases were sampled prior to arrival of the flame front (Figure 1c). A total
of 29 compounds were detected and confirmed using MALT5 and OPUS 5.5. Along with CO,
$CO_2$ and nitrogen compounds, the gas-phase species are largely lightweight hydrocarbons (HCs),
volatile organic compounds (VOCs) and oxygenated volatile organic compounds (OVOCs). Table
2 provides a summary of all compounds observed during the static measurements and is broken
down into subcategories of chemical classes by rows labeled a-e, with ambient gases such as CO
and $CO_2$ in group a, alkanes and alkenes in group b, alcohols, aldehydes and carboxylic acids in
group c, aromatic species in group d, and N-bearing compounds in group e. The benefits of the *in*
*situ* laboratory static measurements were controlled gas sample collection with FTIR analysis and
longer scan times for increased SNRs at higher spectral resolution. Valves were shut before the
flame front arrived allowing for minimal mixing of air and flame gases near the extractive probe.
In this manner the targeted pyrolysis phase was likely to be sampled with a greater mole fraction
rather than that of the combustion phase. The gases listed in Table 2 have previously been
observed in smoke in either field or laboratory settings, and some have been linked to pyrolysis
(Scharko, 2019a,b; Burling et al., 2010, 2011; Christian et al., 2003, 2004; Gilman et al., 2015;
Goode et al., 1999, 2000; Hatch et al., 2017; Selimovic et al., 2018; Stockwell et al., 2014;
Yokelson et al., 1996, 1997; Akagi et al., 2013, 2014; Alves et al., 2010; Hurst et al., 1994a, b;





Karl et al., 2007; Paton-Walsh et al., 2010). Compounds associated with the pyrolysis phase and
observed in several of the static measurements include acetic acid, ethene ($C_2H_4$), allene, 1,3-
butadiene, acetaldehyde, formic acid, formaldehyde, acrolein, benzene, furan, furaldehyde,
naphthalene and phenol.
As seen in Table 2, ammonia gas ($NH_3$) was also detected at fairly low mixing ratios in the
laboratory scale experiments, which had previously not been detected in the Ft. Jackson field
study:  The lack of $NH_3$ detection in those studies was ascribed to the known adsorptivity of the
compound as it may have adhered to either the transfer canister walls, the extractive probe, or the
White Cell, all at ambient temperatures as used in those studies (Scharko et al., 2019; Roscioli et
al., 2015; Stockwell et al, 2014; Yokelson et al., 2003; Neuman et al., 1999). Adhesion losses
were minimized in the present experiments by a) measuring the gas parcel directly without storage
and b) heating transfer lines and gas cell to ~55 °C.
**Table 2.** Mixing ratio of chemicals from spectra collected using the static acquisition method. Burns are labeled
by number and plant species. Compound mixing ratios are reported in ppm (with the exception of $H_2O$ and $CO_2$
reported as percents) and categorized by (a) background ambient compounds, (b) simple hydrocarbons, (c)
oxygenated organic compounds,  (d) aromatics, and (e) N-bearing species.

|   |   | Burn 76 | Burn 78 | Burn 80 | Burn 82 | Burn 84 | Burn 88 | Burn 89 | Burn 90 | Burn 94 | Burn 95 |
|---|---|---|---|---|---|---|---|---|---|---|---|
|   |   | inkberry | sparkleberry | sparkleberry | sparkleberry | fetterbush | fetterbush | inkberry | sparkleberry | sparkleberry | sparkleberry |
| a | % $H_2O$ | 1.24 | 1.05 | 3.23 | 2.03 | 3.54 | 3.08 | 3.46 | 1.82 | 6.21 | 4.10 |
|   | % $CO_2$ | 0.06 | 0.09 | 2.06 | 0.48 | 1.51 | 2.06 | 1.36 | 0.34 | 4.60 | 2.08 |
|   | CO | 1.45 | 3.90 | 808 | 192 | 1089 | 1057 | 391 | 160 | 7506 | 2651 |
|   | $N_2O$ | 0.35 | 0.34 | 1.21 | 0.50 | 1.28 | 1.79 | 0.44 | 0.41 | 3.22 | 1.78 |
| b | $CH_4$ | 2.27 | 2.21 | 45.3 | 10.7 | 54.5 | 50.3 | 15.4 | 11.3 | 682 | 198 |
|   | $C_2H_2$ | 0.01 | 0.06 | 23.8 | 4.52 | 23.4 | 23.2 | 8.82 | 5.62 | 351 | 96.5 |
|   | $C_2H_4$ | 0.07 | 0.05 | 29.3 | 7.05 | 39.9 | 39.3 | 9.66 | 6.52 | 452 | 133 |
|   | $C_2H_6$ |   |   | 0.83 |   |   | 2.76 |   | 4.00E-04 | 24.2 | 6.29 |
|   | $C_3H_6$ |   |   | 4.02 | 0.99 | 5.55 | 5.48 | 0.75 | 0.77 | 61.3 | 18.1 |
|   | allene | 0.17 |   | 0.64 | 0.29 | 1.12 | 1.21 | 0.25 | 0.12 | 8.69 | 2.30 |
|   | 1,3-butadiene |   |   | 1.63 | 0.37 | 1.98 | 2.07 | 0.26 | 0.43 | 28.1 | 7.57 |
|   | isobutene |   |   | 0.75 |   | 0.74 | 0.52 |   |   | 3.16 | 1.07 |
|   | isoprene |   |   | 1.78 | 0.39 | 1.72 | 1.43 | 0.31 | 0.32 | 11.7 | 4.22 |
| c | $CH_3OH$ | 0.89 | 0.24 | 6.81 | 1.53 | 6.92 | 9.44 | 1.66 | 0.93 | 42.3 | 18.0 |
|   | $C_2H_5OH$ | 1.37 |   |   |   |   |   |   |   |   |   |
|   | acetic acid | 0.07 |   | 5.93 | 3.55 | 13.4 | 13.8 | 11.0 | 2.49 | 13.4 | 9.62 |
|   | formic acid |   |   | 15.9 | 5.14 | 32.35 | 35.3 | 9.20 | 3.64 | 130 | 73.6 |
|   | acetaldehyde |   |   | 5.87 | 1.69 | 7.62 | 8.65 | 1.51 | 0.94 | 73.6 | 22.6 |
|   | acrolein |   |   | 2.59 | 1.29 | 3.99 | 4.35 | 0.98 | 0.00 | 26.0 | 9.53 |
|   | crotonalehyde |   |   | 1.51 | 0.54 |   |   | 0.73 | 0.17 | 9.97 | 5.64 |
|   | formaldehyde |   | 0.08 | 13.6 | 4.31 | 21.3 | 22.5 | 5.41 | 3.33 | 114 | 52.8 |
| d | benzene |   |   | 4.08 | 2.23 | 5.19 | 4.24 | 1.93 | 1.48 | 61.3 | 18.6 |
|   | furan |   |   | 0.75 |   | 0.39 | 0.54 |   |   | 3.07 | 1.16 |
|   | furfural |   |   | 0.65 | 0.06 |   |   |   | 0.13 | 3.34 | 1.24 |
|   | naphthalene |   |   | 4.48 | 1.06 | 3.60 | 4.80 | 3.40 | 0.82 | 14.6 | 1.42 |
|   | phenol |   |   | 0.90 | 0.30 | 1.36 | 1.63 | 1.75 | 0.37 | 2.19 | 1.63 |
| e | $NH_3$ | 0.10 | 0.29 | 0.19 | 1.29 | 1.79 | 0.88 | 1.08 | 0.41 | 0.58 | 0.57 |
|   | HCN |   |   | 5.84 | 2.19 | 8.25 | 6.94 | 3.36 | 1.69 | 64.2 | 21.0 |
|   | HNCO |   |   | 1.89 | 0.67 | 2.61 | 2.94 | 1.27 | 0.70 | 5.37 | 1.98 |
|   | HONO |   | 0.11 | 9.40 | 2.53 | 9.72 | 12.7 | 8.92 | 1.75 | 26.9 | 11.3 |





When comparing the RFL laboratory scale experiments to the 2018 Ft. Jackson field scale
experiments, it is evident that field scale values via the static mode are greater than those of the
laboratory, even though the laboratory experiment attempted to replicate Ft. Jackson fuel beds and
scenarios. In most cases, a comparison of compounds found in the RFL laboratory burns and the
Ft. Jackson 2018 field burns finds Ft. Jackson mixing ratios approximately 4 to 10x greater than
those of the RFL 2018 tunnel data. Field scale measurements typically yield more emissions than
experiments conducted in the laboratory (Yokelson, 2013; Scharko, 2019b, Weise et al. 2015).
However, while the mixing ratios may differ or be larger/smaller, the information describing the
composition of the mixture is relative in nature and is contained in log-ratios of the various gases.
But analysis of the data as compositional data (Aitchison 1986) is beyond the scope of the present
paper. Table 3 displays the minimum and maximum mixing ratio values in ppm for five
compounds from the Ft. Jackson studies presented in Scharko et al. (2019a) versus the present
RFL laboratory results.  Of the five species compared, acetaldehyde, acrolein, and allene all follow
the trend of having Ft. Jackson results being significantly higher than the RFL studies by a factor
of ~4. Naphthalene, a polycyclic aromatic hydrocarbon (PAH) was the only exception to this trend,
having comparable mixing ratio values in the two studies. This anomaly could be attributed to one
of naphthalene's pyrolysis formation route as suggested by Fairburn et al., where a single ringed
aromatic compound undergoes a Diels-Alder reaction of an alkene (Fairburn et al., 1990; Liu et
al., 2017).  Of the four compounds compared, naphthalene is the only one to be derived from a
secondary reaction, whereas acetaldehyde and acrolein are derived directly from the pyrolysis of
cellulose (Stein et al., 1983), while allene is a compound known to be a precursor of aromatic
compounds and soot (Frenklach et al., 1983, 1988).  As noted, most compounds detected in the
RFL laboratory studies yielded ~4 to 10x lower mixing ratios compared to the field scale studies
at Ft. Jackson.   Along with naphthalene, however, acetic acid, formaldehyde, isoprene and
isobutene were also found to have comparable mixing ratios to those reported in the Ft. Jackson
studies.   This could be due to the four compounds being products of secondary reactions, or
fragmentation, of species such as lignin, xylan and glucomannan (Collard and Blin, 2014). It
should be noted that of the five novel compounds detected in Scharko et al. (2019b), only four
were detected in these laboratory scale experiments.  Methyl nitrite was not observed (Table 3).
This is attributed to the field experiment being on the Ft. Jackson base where there is known to be



unexploded ordinance (Scharko et al., 2019b) or possibly due to lower concentration levels that
are below the detection limits of the present laboratory-scale experiment.
**Table 3.** Calculated minimum and maximum mixing ratios (ppm) for the 10 canister measurements taken at the
Ft. Jackson field measurements (Scharko et al., 2019) along with the minimum and maximum mixing ratios
(ppm) for the 10 static measurements during the RFL laboratory experiment of acetaldehyde, acrolein, allene,
methyl nitrite and naphthalene.

| Target Compound | Ft. Jackson calculated mixing ratio (ppm) | | | RFL calculated mixing ratio (ppm) | |
|---|---|---|---|---|---|
| | min | max | | min | max |
| acetaldehyde | 34.5 | 264.8 | | 0.94 | 73.6 |
| acrolein | 14.7 | 125.7 | | 0.98 | 26 |
| allene | 2.2 | 37.8 | | 0.12 | 8.69 |
| methyl nitrite | 2.3 | 21 | | - | - |
| naphthalene | 1.4 | 19.9 | | 0.86 | 14.6 |


It is clear that the static method as deployed was not perfect at either isolating strictly the pyrolysis
phase gases or capturing extremely high fractions of combustion gases. The method relied heavily
on valves being closed prior to the flame front using visual cues as opposed to using other
techniques, e.g. thermal IR. While not readily visible to the human eye, radiant and convective
heating (as determined by Background-Oriented Schlieren measurements - Aminfar et al 2019)
occurred well in advance of the flame front, suggesting this as a possible alternate visualization
of pyrolysis gas release (Aminfar 2019).  In any case, there is a narrow temporal window for the
pre-combustion phase making the valve-close time extremely important.  For example, spectra
from burns 76 and 78 show largely the detection of only ambient air compounds indicating the
valves were closed too early.  Conversely, in other samples there is clearly some mixing of both
upstream and downstream air before the gas enters the extractive probe. Despite the shortcomings
of the static method most of the attempts to obtain pre-combustion gases were successful as
evidenced in part by the chemical composition of the isolated gases.

*3.2 Spectral-Thermal Correlation to Isolate Pyrolysis Phase*
Dynamic IR data and visual image acquisition proved advantageous to resolve the different phases
of the experiments (e.g. pyrolysis, flaming combustion, smoldering combustion). This was





important since MCE is a function of the gas composition and is not unique to phase (i.e., the same
value of MCE results if the same relative amounts of CO and $CO_2$ are observed, whether in the
pyrolysis, flaming or smoldering combustion phases) are less appropriate due to the lack of arrival
of the flame front and onset of combustion. MCE, defined as $\Delta CO_2/(\Delta CO + \Delta CO_2)$, has many
times been used to distinguish phases of combustion, namely flaming vs. smoldering although
Ward and Radke (1993) recommended combustion efficiency as the preferred descriptor of the
combustion system. MCE has not been used to identify pyrolysis nor should it be for the non-
uniqueness described previously. Recent studies have introduced more sophisticated techniques to
analyze smoke emissions data with compositional data methods (Weise et al. 2020). However,
since primary and secondary pyrolysis occurs both prior to and after the onset of combustion or
oxidation, methods such as the MCE are not appropriate. We were not able to use the metric
suggested by Sekimoto et al. (2018), namely high temperature vs. low temperature pyrolysis as
determined from the acetylene-to-furan ratio due to weak furan signals in the present study due to
shortened scan times. The analysis was further exacerbated because furan's strongest vibrational
band, the $\nu_{19}$ vibrational band near 745 $cm^{-1}$, corresponding to the C-H out-of-plane bend
(Shimanouchi, 1972), was obscured by saturated carbon dioxide lines and thus MALT was not
able to generate a satisfactory fit for this microwindow.
The pre-flame arrival gases were identified by either of two methods: The first method involved a
simple time subtraction of 4.8 s from the recording of the infrared spectrum time stamp and
associating that time to the corresponding visual and FLIR thermal infrared video images (Fig. 3).
This provided a relatively accurate verification that the gases being investigated were emitted prior
to the onset of combustion as seen in Table 4. The second method used the FTIR spectra directly:
demarcations for the flame front were denoted by the maximal value obtained for both CO and
$CO_2$ concentrations, i.e. greatest fraction of gas from the combustion phase. From this value the
FTIR's scans were selected for pyrolysis corresponding to the ~10 seconds before arrival (~0.1 m
distance) of the flame front.

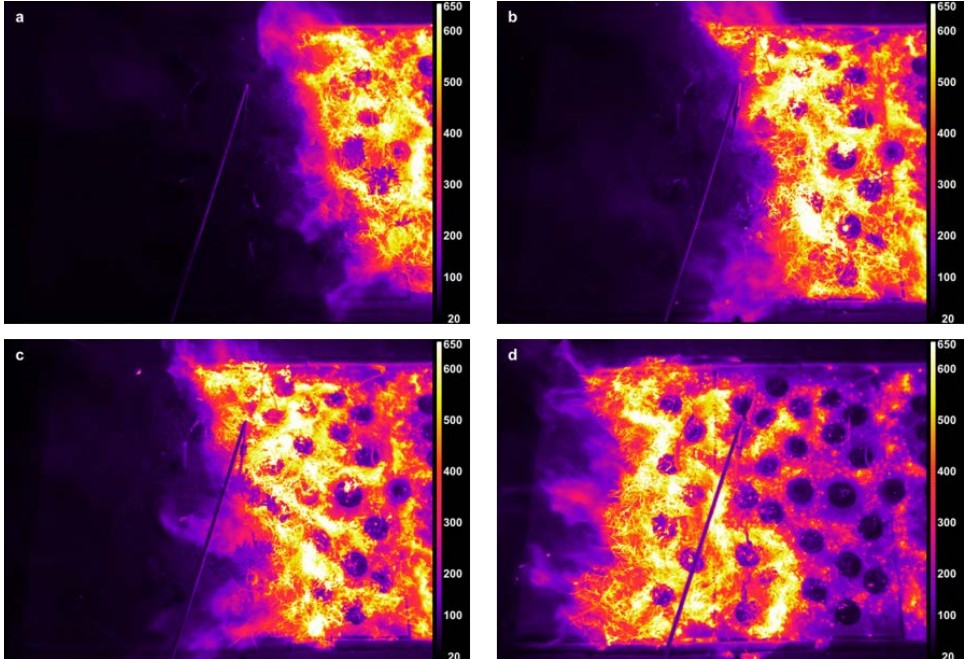

**Figure 3.** Burn 87 inkberry on longleaf pine needle fuel bed – FLIR thermal imaging for burn progression. (a) frame corresponding to FTIR scan 5 signaling the pre-combustion phase, (b) frame corresponding to FTIR scan 16, flame front nearing sample probe, (c) frame corresponding to FTIR scan 21, inkberry bush consumed by flame, (d) frame corresponding to FTIR scan 44, flame front has passed the probe. Dark circles are ceramic plant holders.

The FTIR time-resolved scans (including derived chemical mixing ratios) synchronized to the RFL time-stamped thermal IR temperature images provide insight into the chemical composition of each burn. As an example, Table 4 pairs data from the two systems for Burn 87. FTIR scan number, FTIR time stamp, RFL FLIR recorded temperature near the extractive probe, and a selection of chemical concentrations are shown. The table demonstrates that spectral data for FTIR scans 0-8 saw no significant detections above ambient levels as corroborated by the FLIR images displaying temperatures range from 40-80 °C (see Table 4 and Figure 3a); the extractive probe is still in the low temperature region. The gradual increase in mixing ratios for most compounds (excluding ammonia, which is primarily a smoldering gas) begins after FTIR scan 9. The magenta and orange colored domains seen in Figure 3b indicate the encroaching flame front and a rise in thermal temperatures. The frames corresponding to FTIR scans 16-19 display IR temperatures between 175 and 220 °C. In this temperature range compounds associated with the pyrolysis phase such as acetaldehyde, acetic acid and allene (shown in Table 4) are not only manifest in the IR spectra, but their mixing ratios rise rapidly. Shortly thereafter the greatest mixing ratios of $CO_2$ occur at scans


20 through 22, indicating the flaming stage; this is corroborated by thermal IR video of the inkberry
plant beginning to be fully consumed in flames (Fig 3c). As the flame front progressed down the
tunnel, temperatures near the plant holder began to drop with the onset of the smoldering phase as
indicated by lower mixing ratios as well as the thermal IR visual seen in Fig. 3d. [We note in
Figure 3 that the temperature directly near/above the holders is much cooler due to minimal duff
cover and the plants being green.] The video stopped recording at scan 48, when the flame reached
the end of the fuel bed although the FTIR continued to collect interferograms to monitor
smoldering from the fire.

**Table 4:** Burn 87 inkberry amongst pine needle fuel bed FTIR scan summary synchronized to FLIR temperature data.
Scan number, FTIR time stamp, along with FLIR video emissions temperature at extractive probe accounting for time
delay and mixing ratios from carbon dioxide ($CO_2$), carbon monoxide (CO), ethene ($C_2H_4$), acetic acid ($CH_3COOH$),
formaldehyde (HCHO), acetaldehyde ($CH_3CHO$), and phenol ($C_6H_6O$).

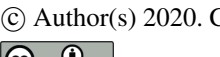



| | FTIR Scan Number | FTIR time stamp | 4.8 s Delayed FLIR Video Time Stamp | FLIR temperature at inlet (°C) | $CO_2$ (ppm) | CO (ppm) | $C_2H_4$ (ppm) | $CH_3COOH$ (ppm) | HCHO (ppm) | $CH_3CHO$ (ppm) | $C_6H_6O$ (ppm) |
|---|---|---|---|---|---|---|---|---|---|---|---|
| **Ambient** | scan 0 | 10:41:23.69 | 10:41:18.89 | 41.4 | 1548 | 49.5 | 2.4 | 0.0 | 0.6 | - | - |
| | scan 1 | 10:41:25.20 | 10:41:20.40 | 44.2 | 1912 | 74.5 | 1.9 | 1.7 | 0.9 | - | - |
| | scan 2 | 10:41:26.70 | 10:41:21.90 | 51.0 | 1562 | 63.7 | 2.3 | 1.1 | 0.3 | - | - |
| | scan 3 | 10:41:28.21 | 10:41:23.41 | 54.9 | 1290 | 58.4 | 1.1 | 0.5 | 0.5 | - | - |
| | scan 4 | 10:41:29.71 | 10:41:24.91 | 53.4 | 1882 | 63.1 | 1.2 | 1.2 | 0.8 | - | - |
| | scan 5 | 10:41:31.21 | 10:41:26.41 | 63.2 | 1946 | 57.0 | 1.5 | 1.1 | 0.4 | - | - |
| | scan 6 | 10:41:32.72 | 10:41:27.92 | 72.8 | 3811 | 102 | 4.7 | 1.8 | 1.2 | - | - |
| | scan 7 | 10:41:34.22 | 10:41:29.42 | 77.6 | 4722 | 138 | 2.7 | 1.9 | 1.5 | - | - |
| | scan 8 | 10:41:35.73 | 10:41:30.93 | 86.2 | 3553 | 109 | 0.9 | 2.2 | 1.4 | - | - |
| **Volatilization + Pyrolysis** | scan 9 | 10:41:37.23 | 10:41:32.43 | 116.9 | 2957 | 97 | 1.0 | 2.4 | 0.7 | - | - |
| | scan 10 | 10:41:38.73 | 10:41:33.93 | 132.1 | 3360 | 138 | 1.8 | 2.8 | 1.2 | 1.4 | - |
| | scan 11 | 10:41:40.24 | 10:41:35.44 | 115.7 | 7476 | 246 | 4.2 | 2.6 | 3.6 | 0.0 | - |
| | scan 12 | 10:41:41.74 | 10:41:36.94 | 121.3 | 10274 | 291 | 6.2 | 9.0 | 4.8 | 4.2 | 0.7 |
| | scan 13 | 10:41:43.25 | 10:41:38.45 | 183.3 | 10890 | 391 | 11.4 | 11.5 | 7.1 | 1.1 | 1.6 |
| | scan 14 | 10:41:44.75 | 10:41:39.95 | 159.4 | 11833 | 635 | 23.5 | 14.5 | 13.5 | 3.3 | 1.5 |
| | scan 15 | 10:41:46.25 | 10:41:41.45 | 163.8 | 16080 | 1214 | 48.5 | 16.8 | 28.7 | 7.2 | 1.5 |
| | scan 16 | 10:41:47.76 | 10:41:42.96 | 176.9 | 25757 | 2217 | 95.3 | 16.3 | 54.4 | 17.6 | 1.0 |
| | scan 17 | 10:41:49.26 | 10:41:44.46 | 181.3 | 31856 | 2915 | 129 | 14.4 | 70.5 | 24.0 | 1.4 |
| | scan 18 | 10:41:50.77 | 10:41:45.97 | 220.1 | 41291 | 2878 | 260 | 13.1 | 92.6 | 34.5 | 1.4 |
| | scan 19 | 10:41:52.27 | 10:41:47.47 | 219.0 | 61166 | 8228 | 435 | 12.3 | 121 | 44.3 | 2.1 |
| **Flaming Combustion** | scan 20 | 10:41:53.77 | 10:41:48.97 | 296.1 | 79332 | 11354 | 747 | 13.1 | 178 | 80.7 | 2.9 |
| | scan 21 | 10:41:55.28 | 10:41:50.48 | 261.7 | 54381 | 9729 | 1167 | 17.6 | 255 | 140 | 4.7 |
| | scan 22 | 10:41:56.78 | 10:41:51.98 | 456.1 | 64077 | 12954 | 1025 | 15.3 | 185 | 103 | 6.0 |
| | scan 23 | 10:41:58.29 | 10:41:53.49 | 429.9 | 41495 | 8620 | 530 | 15.4 | 123 | 63.3 | 6.3 |
| | scan 24 | 10:41:59.79 | 10:41:54.99 | 516.5 | 25879 | 3453 | 257 | 15.5 | 74.5 | 33.9 | 6.8 |
| | scan 25 | 10:42:01.29 | 10:41:56.49 | 514.5 | 15965 | 3110 | 116 | 15.1 | 45.7 | 19.6 | 6.8 |
| | scan 26 | 10:42:02.80 | 10:41:58.00 | 460.1 | 11819 | 2416 | 53.2 | 15.6 | 35.2 | 12.0 | 5.6 |
| | scan 27 | 10:42:04.30 | 10:41:59.50 | 453.7 | 8566 | 1875 | 36.6 | 14.7 | 27.4 | 6.9 | 5.2 |
| | scan 28 | 10:42:05.81 | 10:42:01.01 | 448.0 | 5795 | 1320 | 14.5 | 13.2 | 21.8 | 3.0 | 5.6 |
| | scan 29 | 10:42:07.31 | 10:42:02.51 | 440.2 | 5235 | 1302 | 11.0 | 14.6 | 20.0 | 5.1 | 4.5 |
| | scan 30 | 10:42:08.81 | 10:42:04.01 | 484.7 | 3626 | 916 | 5.8 | 15.1 | 15.0 | 5.2 | 4.2 |
| | scan 31 | 10:42:10.32 | 10:42:05.52 | 470.4 | 2368 | 570 | 3.3 | 11.5 | 11.1 | 2.7 | 3.8 |
| | scan 32 | 10:42:11.82 | 10:42:07.02 | 497.5 | 1636 | 377 | 1.6 | 10.8 | 9.4 | 0.3 | 3.7 |
| | scan 33 | 10:42:13.33 | 10:42:08.53 | 477.4 | 1684 | 399 | 1.0 | 9.2 | 8.8 | 2.1 | 3.4 |
| | scan 34 | 10:42:14.83 | 10:42:10.03 | 450.1 | 1986 | 519 | 0.9 | 10.4 | 9.5 | -1.5 | 3.0 |
| **Smoldering Combustion** | scan 35 | 10:42:16.33 | 10:42:11.53 | 397.9 | 1968 | 518 | 0.7 | 9.9 | 9.3 | 2.8 | 3.2 |
| | scan 36 | 10:42:17.84 | 10:42:13.04 | 410.2 | 1901 | 495 | 1.4 | 9.6 | 8.8 | 0.9 | 2.8 |
| | scan 37 | 10:42:19.34 | 10:42:14.54 | 401.6 | 1936 | 516 | 1.5 | 9.2 | 9.4 | 0.0 | 2.7 |
| | scan 38 | 10:42:20.85 | 10:42:16.05 | 358.3 | 1935 | 513 | 1.6 | 9.0 | 9.5 | 1.1 | 2.8 |
| | scan 39 | 10:42:22.35 | 10:42:17.55 | 341.8 | 1753 | 439 | 1.8 | 9.7 | 8.9 | 1.1 | 2.3 |
| | scan 40 | 10:42:23.85 | 10:42:19.05 | 320.6 | 1438 | 345 | 1.4 | 10.4 | 8.3 | -0.5 | 2.9 |
| | scan 41 | 10:42:25.36 | 10:42:20.56 | 305.9 | 1224 | 277 | -0.1 | 10.9 | 7.4 | 1.4 | 2.8 |
| | scan 42 | 10:42:26.86 | 10:42:22.06 | 295.0 | 1377 | 324 | 1.4 | 11.6 | 8.2 | -2.0 | 2.4 |
| | scan 43 | 10:42:28.37 | 10:42:23.57 | 272.5 | 1629 | 411 | 1.1 | 11.8 | 8.4 | 0.3 | 2.4 |
| | scan 44 | 10:42:29.87 | 10:42:25.07 | 258.3 | 1366 | 325 | 0.8 | 12.8 | 7.8 | 2.5 | 2.7 |
| | scan 45 | 10:42:31.37 | 10:42:26.57 | 260.1 | 1059 | 212 | 0.4 | 11.9 | 6.7 | 3.4 | 2.7 |
| | scan 46 | 10:42:32.88 | 10:42:28.08 | 238.5 | 1037 | 212 | 0.8 | 13.1 | 7.1 | 0.5 | 2.3 |
| | scan 47 | 10:42:34.38 | 10:42:29.58 | 223.9 | 1094 | 236 | 0.9 | 14.4 | 6.7 | 1.1 | 1.2 |
| | scan 48 | 10:42:35.89 | 10:42:31.09 | 226.7 | 1117 | 248 | 1.2 | 13.1 | 7.0 | -0.4 | 2.7 |



As stated, a second method was also used to analyze/corroborate the different stages of the burn,
whereby mixing ratios of $CO_2$, CO and $C_2H_4$ were analyzed to find their burn maxima (Viatte et
al., 2015). The $CO_2$ elevated mixing ratios (esp. relative to CO) are associated with the hottest,





flaming stage of biomass burns (Yokelson et al., 1996). To temporally isolate the flaming stage,
the MCE criteria was employed and values of 88-95% indicative of smoldering were found for the
region. Having identified the flaming stage, the pyrolysis stage was estimated by subtracting 6-8
seconds from that spectrum with maximal $CO/CO_2$ emissions, corresponding to ~4 FTIR scans (at
1 cm$^{-1}$ resolution).  The agreement between the two methods was quite good and helped to
demarcate the stages as seen in Table 4.

Figure 4 displays the infrared spectral progression of Burn 87, longleaf pine needles with inkberry,
bed at 1.0 cm$^{-1}$ resolution looking at two different spectral regions. The CO (and $CO_2$) profiles are
seen in Fig. 4a. Noted on the z-axis is scan 22; scans 20-22 are the time frames where maximal
$CO_2$ and CO emissions were observed; the region is also denoted by red spectral traces. Once the
flaming stage had been identified, the pyrolysis phase was then demarcated; in the pyrolysis phase
CO was evident (partially from upwind mixing) and was beginning to significantly increase; the
stage is indicated by orange traces (scans 16-19) in Figure 4. Other stages assigned were noted as
the pre-flame stage where $\Delta CO$ and $\Delta CO_2$ were near zero in the FTIR data and are seen as scans
0-8 with purple traces.  Blue traces correspond to the smoldering phase of combustion, where $CO_2$
mixing ratios decreased, the flame front had passed the extractive probe and MCE values were on
the order of 85-75%. The spectral profile and mixing ratios of ethene ($C_2H_4$) were also used to
evaluate the time-resolved FTIR data. (Johnson et al. 1993) This lightweight hydrocarbon is a
product of primary pyrolysis and if detected can be used to determine certain stages of the burn
(e.g. Yang et al., 2007). Figure 4b displays primarily the $\nu_7$ band of ethene at 949.4 cm$^{-1}$
(Shimanouchi, 1972).  Ethene reached its maxima mixing ratio at scan 22 (red traces) before it
quickly disappeared, being a pyrolysis gas that was oxidized by the flame. It was first seen to
appear as early as scan 13 (green traces) but became clearly evident in scan 16 (orange traces,
pyrolysis phase) and continued to grow. The rapid disappearance of $C_2H_4$ upon combustion is
similar to that of formaldehyde and acetaldehyde (Table 4) whose concentrations also dropped
after scan 23, but the disappearance is juxtaposed with acetic acid whose values remained
~constant throughout the flaming and smoldering phases. As seen in the IR data, the $C_2H_4$ gas
signal corroborated that ethene is a key product of the primary pyrolysis phase.  Other compounds
showing significant signals in this time domain and described as pyrolysis gases include acrolein
and allene. (Scharko 2019a; Akagi et al., 2013; Frenklach et al. 1983, 1988; Stein et al., 1983;
Koss et al., 2018; Brilli et al., 2014).

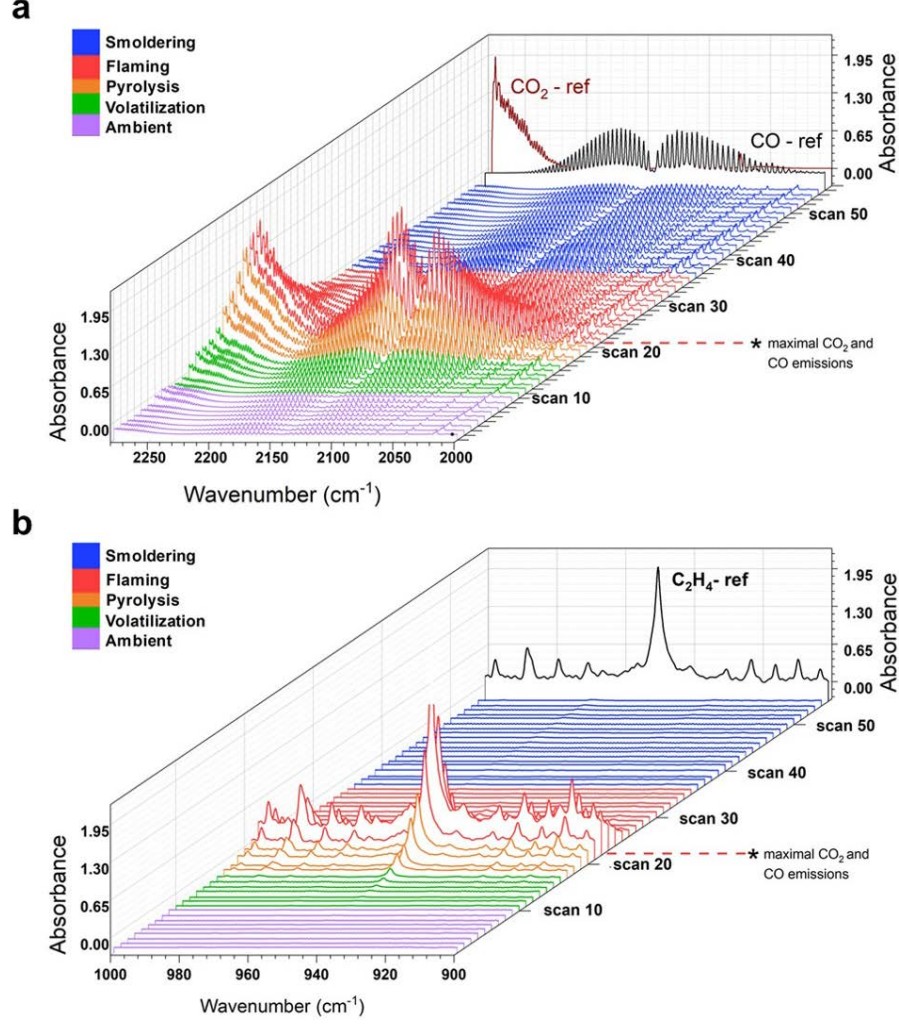


**Figure 4.** Burn 87, inkberry with longleaf pine needles: a) CO and $CO_2$ spectral profile from 2250-2000 cm$^{-1}$.
Purple traces indicate the ambient stage, green and orange traces indicate the pre-combustion/pyrolysis stage,
red spectral traces indicate the flaming stage and blue traces indicate smoldering. b) Largely $C_2H_4$ spectral
waterfall plot from 1000-900 cm$^{-1}$ with accompanying $C_2H_4$ reference spectrum as black trace.

The two methods to determine the pyrolysis, flaming, combustion and smoldering phases yielded
congruous results: The isolated burn stages determined from method one, in which FTIR gas-
phase data were synchronized to the FLIR thermal imaging, and from method two, using the FTIR
time-resolved data only, were found to be virtually identical. This is evidenced by linking the scans



determined to be in the pyrolysis phase (scans 16-19) using method two as seen in Fig. 4, with the
temperature data recorded by the FLIR using method one and seen in Table 4.  For these scans,
the temperature profile ranges from 175-220 °C, corresponding to temperatures associated with
first stages of pyrolysis.
For most analytes biomass burning gas mixing ratios, the concentration values observed at the
peak of the dynamic measurements values were significantly greater than for the concentrations
recorded in the static measurements. The dynamic experiments were of course carried out for the
duration of the burn, whereas the static burns (in an effort to characterize pre-combustion phases)
attempted to isolate a specific time when the pyrolyzate concentrations were maximized.
Analyzing the data using the dynamic technique allowed for confirmation of certain compounds
such as naphthalene, allene, acetaldehyde, and acrolein as compounds that appeared during the
pyrolysis phase. These compounds, which have been previously detected as pyrolysis gases using
FTIR for field plot burns, (Scharko et al. 2019) were again observed during these laboratory scale
tests and in almost all cases appeared before the flame front encroached on the sampling probe.

*3.3 Dynamic Detection of Phenol in the Pre-combustion Through Smoldering Stages*

In the present study phenol ($C_6H_6O$) was detected during several burns; its origin ascribed to the
pyrolysis of lignin(s) (Kibet et al. 2012, Hawthorne et al. 1989) and has been mostly observed
using other techniques such as gas chromatography mass spectrometry (GC-MS). (Saiz-Jimenez
et al. 1986) Phenol and phenolic compounds are also known to contribute to the formation of
secondary organic aerosols. (Yee et al., 2013) It has been observed in simple pyrolysis
experiments, emanating from both pine and spruce species (e.g. Saiz-Jimenez and De Leeuw,
1986, Ingemarsson et al., 1998). In addition to simply pyrolytic emissions, phenol has also been
identified as a common component of tar as a pyrolysis product.  In biomass burning, phenol has
been observed using both FTIR and other methods, (Gilman et al. 2015, Yokelson et al. 2013), e.g.
proton-transfer mass spectrometry (PTR-MS) and GC-MS. In 2013 phenol was detected in a
closed cell, airborne FTIR field experiment but not in an open-path FTIR lab experiment
(Yokelson et al. 2013). The absence of $C_6H_6O$ in the lab experiment was attributed to the lack of
consumption of rotten wood as fuel.  In those studies, airborne phenol emissions measured in the
field with closed-cell FTIR were also noted as being 2 to 4x greater than the phenol emissions
captured by PTR-MS in the laboratory.





In the present experiments, phenol was detected in 8 of the 10 static measurements (recall that two
of the static measurements only showed ambient gases due to early closure of the valves). Figure
5 demonstrates the static spectrum from Burn 89, corresponding to the burning of longleaf pine
with inkberry. Seen in Fig. 5 are the experimental spectrum (blue trace) and also the reference
spectrum of acetic acid (green trace). After subtraction of the $CH_3COOH$ vapor spectrum, the
residual contained two small peaks which were readily identified as phenol vapor via the $\nu_{15}$
vibrational band near 1176.2 cm$^{-1}$, as well as the $\nu_{16}$ band at 1150.2 cm$^{-1}$ (Keresztury et al., 1998).
The phenol reference spectrum from the PNNL spectral library (red trace) was then subtracted
from that residual (purple trace) with an overall residual that is mostly noise (black trace). [For the
dynamic spectra the process is repeated for each of the individual spectral time slices, represented
by scan number using the concentration of phenol determined by the MALT program.] To confirm
the spectral analysis, in each case the mixing ratio calculated by MALT was converted to a
spectrum by multiplying by the appropriate concentration path length factor; the predicted
spectrum was visually compared to the actual data.

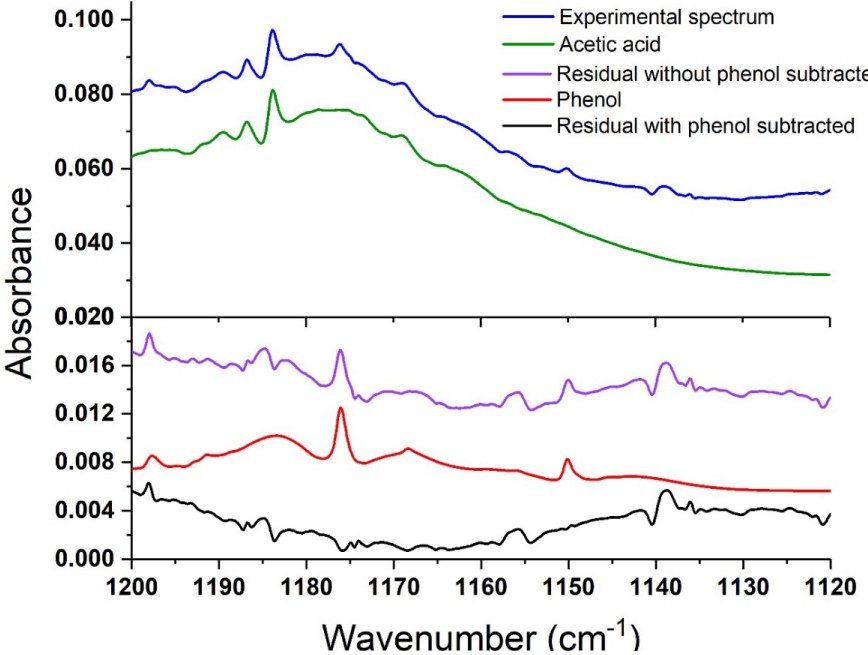


**Figure 5.** Static spectrum obtained from Burn 89 (1 kg longleaf pine needles with inkberry). The blue trace
is the FTIR experimental spectrum, the green trace the reference spectrum of acetic acid, the purple trace
the residual after acetic acid subtraction, the red trace the reference spectrum of phenol and the black trace
residual after phenol optimization/subtraction.





Phenol was also detected using the dynamic method and Figure 6 displays a series of dynamic
spectra recorded for Burn 87. The spectra in the left frame (a) are individual spectra after the acetic
acid (CH$_3$COOH) spectral component has been subtracted from the spectrum for each time slice,
all recorded at 1.0 cm$^{-1}$ resolution with 54 total measurements recorded at Δt=1.5 seconds.  While
the spectral noise is still significant, the presence of phenol peaks, particularly the ν$_{16}$ Q-branch at
1176.5 cm$^{-1}$ and the ν$_{15}$ peak at 1150.2 cm$^{-1}$, are evident. Optimization for the phenol mixing ratio
in each spectrum allowed for its calculation in individual time slices and the derived phenol-only
spectra are presented as a waterfall plot in the right frame (b). The first clear evidence of phenol is
seen in scans 14 to 18, before reaching a maximum concentration of 6.9 ppm in scan 24; this is
observed in the right frame of Figure 6, coinciding approximately with maximal CO$_2$ concentration
(scan 22), indicating the greatest ratio of smoke/ambient air in the gas cell.

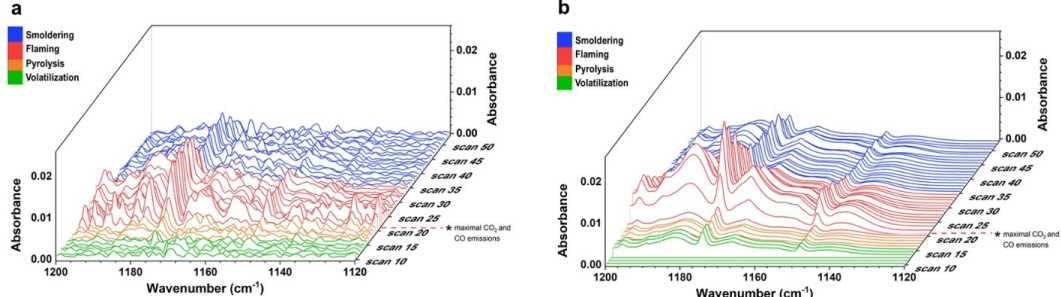


**Figure 6.** Burn 87 –longleaf pine needles with inkberry fuel bed during dynamic mode. Measured and scaled burn spectra showing the progression of phenol during the time resolved study. Acetic acid and water spectral features have been removed in Frame a) with the phenol-only derived mixing ratio spectra in Frame b.

Figure 6 displays the rapid increase of phenol vapor due to the approaching flame front from scan
14 (t = 22.5 s) to its maximal mixing ratio in scan 24 (t = 36 s) followed by a longer gradual phenol
decay with time. This can be juxtaposed with the ethene mixing ratios (seen in Figure 4) that fall
to nearly zero with the onset of combustion; the ethene is consumed by the flame propagation.
Prior to scan 14 in Figure 6b, minimal phenol is observed relative to the noise level and are thus
fit as zero concentration.  Phenol contributions for scans 16-19 can be associated with the pyrolysis
phase of the burn and not combustion. Phenol is one of the major products of 1,2-benzenediol
pyrolysis with maximum yield reported at 800 °C (Ledesma et al. 2002; Thomas et al. 2007). It is
important to note that both temperature and rate of heating influence the composition and yield of
pyrolysis products. Evidenced in Figure 6, the thermal imaging associated with FTIR scan 17



shows a temperature of ~200 °C which is indicated in Kibet et al. (2012) to be within the
temperature range of pyrolysis of lignin: 200 to 400 °C.  Shortly thereafter, phenol mixing ratios
rapidly increase and reach a maximal mixing ratio of 6.9 ppm at scan 24. At scan 24 the flame
front has already reached the extractive probe and thus the maximum intake of smoke and ambient
air is achieved; temperature of the fuel bed is ca. 600 °C, consistent with the flaming phase. The
gradual decay in phenol production as the flame front passes could be due to several factors: (i) an
increased temperature required for complete combustion of the $C_6H_6O$ (ii) residence time of
phenol, (iii) phenol production in the smoldering phase as a tar/char, (iv) adsorption to walls of the
stainless steel tubing and cell.  A cross-section of Figure 6 shows the rapid onset of phenol
production at the temperatures followed by a gradual decay in concentration: this is indicative of
phenol production throughout the burning of inkberry as a species.

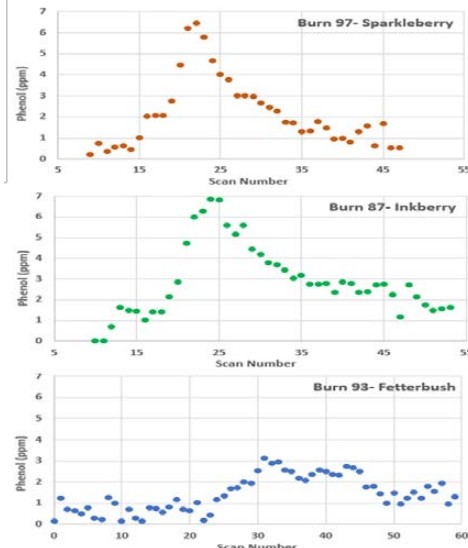


**Figure 7.** Temporal mixing ratios of phenol for different shrub species. Phenol mixing ratios plotted over
time indicated by scan number for Burn 97 sparkleberry (red), Burn 87 inkberry (green), and Burn 20
fetterbush (blue) all on longleaf pine straw bed.

The shape of the temporal profile yields information as to the production of phenol throughout the
evolution of burning. Figure 7 shows the progression of phenol concentration following its first
observed presence in the burn. These graphs are effectively a cross-section of Figure 6, showing
the progression of the height of the phenol peak (directly correlated to phenol concentration)
throughout the burn (with time being represented by scan number in both cases). The level of
phenol generation was observed to vary between plant species. Temporal profiles of phenol
concentration were constructed for burns with three different species: sparkleberry, inkberry, and
fetterbush. These plots illustrate a range of behavior with inkberry and sparkleberry having similar
temporal profiles and similar maxima of ca. 6.5 ppm and fetterbush having a different temporal
profile.  It is important to note that Burn 97 was measured at 0.6 cm$^{-1}$, while burns 87 and 93 were
measured at 1.0 cm$^{-1}$, although the profile of burn 97 is consistent with that of burn 87. [We do not
believe the small change in resolution affects the recovered mixing ratios.] Demonstrated in Figure
7, trace amounts of phenol appear at the onset of combustion and throughout the pyrolysis phase.
Phenol reaches its highest concentrations, however, during the flaming stage as all three temporal
profiles reach maximum during the latter stages of the burn. Moreover, phenol remains throughout
the duration of the burn and is not consumed by secondary reactions, as is e.g. ethene. For these
three burns, fetterbush was observed to have the lowest maximum concentration of phenol, 3.1
ppm, of the three species, while sparkleberry and inkberry had similar maxima (as well as similar
temporal profiles).
The observed differences of phenol in both temporal profile and overall peak concentrations could
arise due to differences in leaf structure and shape, or possibly due to differences in leaf/plant
composition. Pyrolytic production of phenol has been previously attributed to multiple
components of plant composition, including phenol content, lignin content and the amount of
cellulose in each plant species. Therefore, varying phenols, lignin, and cellulose in these plant
species could be the source of phenol concentration variability for each burn.  The physical
composition of multiple plant species, including inkberry and fetterbush was analyzed by Matt and
Dietenberger (2020); it was shown that inkberry has 2.6 times the percentage of phenol by
composition (9.0%) than fetterbush (3.4%)  Although sparkleberry was not included in that study,
it can be suggested that the compositions of inkberry and sparkleberry are similar due to observed
phenol in this experiment as well as plant characteristics. Sparkleberry, a member of the *Vaccinium*
genus, contains many species collectively known as blueberries which are known to contain high
levels of phenolic compounds in the fruits (e.g. Prior et al 1998). This study and the results of Matt
and Dietenberger support the present hypothesis that peak concentrations of phenol are highest for
sparkleberry and inkberry due to higher phenolic content in the plants.
**4.  Summary**



The analytical methods used in this study attempt to provide a detailed view of prescribed burning by enlisting two different FTIR acquisition modes, static and dynamic. By capturing a "snapshot" of a single burn experiment, used in the static method, one can discern the gases with higher specificity and in turn decipher complex spectra by use of chemometrics to extract compounds with high concentrations leaving behind a residual to be analyzed. Lower resolution may hinder these efforts and allow compounds that are present at lower mixing ratios to be obscured by higher absorbing compounds, e.g., carbon dioxide, water, and ethene. In this study we were able to detect additional compounds e.g. phenol, benzene, and allene with greater confidence. However, in gaining specificity there is a loss of time resolution and this is where the dynamic method becomes advantageous. The FTIR dynamic acquisition method when synchronized to thermal imaging, while lower in sensitivity, allows for an overall profile of the burn and can help assign phases to the dynamic stages of the flame. That is, the dynamic method in conjunction with thermal IR imaging provides a more detailed description as temperature and chemical composition profiles can be correlated and assigned to certain phases of the burns. In this study pyrolysis, flaming and smoldering combustion were identified using these new techniques which can aide in the improvement of fire behavior models used by land managers to conduct prescribed fires.

*Data availability.* Data are not publicly available as data release has not been authorized by sponsor of this research.

*Competing interests.* No competing interests.

*Acknowledgements.* This work was supported by the Department of Defense's Strategic Environmental Research and Development Program (SERDP) within project RC-2640, and we gratefully acknowledge our sponsors for their support. PNNL is operated for the U.S. Department of Energy by the Battelle Memorial Institute under contract DE-AC06076RLO 1830.





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
