# Peer review of "Dynamic Infrared Gas Analysis from Longleaf Pine Fuelbeds Burned in a Wind"

_Atmospheric Measurement Techniques, 2020_

## Referee Comment (RC1) · Anonymous Referee #1 · 20 Oct 2020

This manuscript described FT-IR measurements of gases evolved by plant material in various stages of combustion. Bothe the experimental measurements and data analysis were carefully performed, and I recommend that this manuscript be accepted with no changes. Just to show that I did read the manuscript, I did find one minor error. The singular form of "criteria" is "criterion" (line 2 of page 19). Although the following comment need not be addressed in the final version of this paper, I was quite surprised that acetylene was not observed. Was this because the strong Q branch of the bending mode absorbs below the cut-off of the MCT detector or was acetylene simply absent?

[Figure]

Was there no trace of the C-H stretch at ca. 3300 cm-1?

---

## Author Comment (AC1) · 23 Oct 2020

We thank the referee for her/his comments and in the revised version shall change "criteria" to the singular form of "criterion" (line 2 of page 19). To address the other comment, acetylene was in fact observed, in two different spectral regions and it is reported in the Table. We did not call out much attention to it, however, as the focus of the paper changed more from reporting gas phase mixing ratios to reporting on a new technique for atmospheric measurements. We may add a little more text regarding the gas phase findings. Thank you again.

---

## Referee Comment (RC2) · Anonymous Referee #2 · 29 Oct 2020

General Comment

This work focused on gases emissions, and in particular volatile organic compounds (VOCs), by plants materials burned in a wind tunnel, simulating in laboratory a field scenario. The experiments allowed isolating and characterizing pre-combustion phase (pyrolysis) with its specific VOCs sign employing two different methods (FTIR and IR thermal imaging), to validate findings. Moreover, the other fire phases and phenol temporal profile were characterized. Topic is of relevant interest, characterizing an always more spread process in the world, focusing on VOCs emissions, which are

increasingly key factor in atmosphere chemistry and dynamics worldwide. Manuscript is qualitatively satisfactory, fitting journal topic and carrying new useful knowledges to the scientific community. I only have some doubts about the experimental/technical part about VOCs sampling system, based on my experience in VOCs experimental campaigns. I am going to report these perplexities in the specific detailed comments below. However, I recommend accepting this manuscript with some minor revision in structure and specific observation for the technical part.

Specific Comments

1. Line 60, 73. Typing error. There is a dot and then brackets with references and another dot after.

2. From line 72 to 128. There is too much space to explain the entire project into the introduction, respect to the specific goals of the paper. I would summarize the details of the whole project. Indeed, at a first reading it was little bit confusing for me, because I did not find connection in the results.

3. Line 136. I would specify '1 m s-1 wind condition'. Could be confusing.

4. Line 149. I do not know how FTIR Spectrometer works in details, but based on my experience in VOCs measurements with PTR-TOF-MS and cartridges, the best option for VOCs sampling is to use PTFE (Teflon) for sampling line. This is because it is the most inert and least reactive material, avoiding the loss of the sticky compounds as many VOCs could be. It is true that the high temperatures of the gas inside the probe mitigates sticking of compounds on probe walls (showing an average gas temperature inside the probe could be useful, it is available), but I would insert few lines that would take into account consequences of using a stainless steel probe. Because of the stickiness, some compounds could be lost, they could react and become something slightly different from what is primarily emitted by pyrolysis and combustion, or they could be underestimated. This concept is valid in the same way also for the White cell. This is noticed at line 279 for the ammonia in the results.

5. Line 154-156 and Table 1. Why if in the manuscript are reported only the 21 experiments carried out in November 2018, at the beginning of the paragraph all measurement were reported? This could be confusing. I would mention only the experiments showed in the paper.

6. All Chapter 2. Experimental. Always based on my experience in VOCs sampling, I wonder how you took into account the possible contribution to VOCs identification and quantification of the Wind Tunnel, white cell and other canisters? Some compounds could be already present because released by one of these sources and not from the processes that you are surveying in this work, or both could emit them and bias your quantitative estimation. In my experience it is always needed a blank (zero) measurement of the surveying matrix and means. This is important also for the ceramic plant holders. What they emit? What they emit when they burn? This could bias your results.

7. Table 2. I would report standard deviation, since reported mixing ration come from multiple scan averaging of 30 minutes. In this way, it is possible to observe the mixing ration variation during process observation.

8. Line 466-480, paragraph 3.3 . This is a state of art about phenol emissions by burns. It should stay in introduction defining the background knowledges at the base of this study.

---

## Author Comment (AC2) · 22 Dec 2020

Response to Interactive comment on "Dynamic Infrared Gas Analysis from Longleaf Pine Fuelbeds Burned in a Wind Tunnel: Observation of Phenol in Pyrolysis and Combustion Phases" by C. A. Banach et al.

Anonymous Referee #2 General Comment Referee Comments: This work focused on gases emissions, and in particular volatile organic compounds (VOCs), by plants materials burned in a wind tunnel, simulating

in laboratory a field scenario. The experiments allowed isolating and characterizing pre-combustion phase (pyrolysis) with its specific VOCs sign employing two different methods (FTIR and IR thermal imaging), to validate findings. Moreover, the other fire phases and phenol temporal profiles were characterized. Topic is of relevant interest, characterizing an always more spread process in the world, focusing on VOCs emissions, which are increasingly key factor in atmosphere chemistry and dynamics worldwide. Manuscript is qualitatively satisfactory, fitting journal topic and carrying new useful knowledges to the scientific community. I only have some doubts about the experimental/technical part about VOCs sampling system, based on my experience in VOCs experimental campaigns. I am going to report these perplexities in the specific detailed comments below. However, I recommend accepting this manuscript with some minor revision in structure and specific observation for the technical part. Author response We thank the referee the constructive suggestions to improve the readability, utility and strength of the paper. We will try to address the concerns regarding the VOC sampling system in the responses below. Thank you again.

Specific Comments Referee Comments: 1. Line 60, 73. Typing error. There is a dot and then brackets with references and another dot after. Author response Corrected. Thank you.

Referee Comments: 2. From line 72 to 128. There is too much space to explain the entire project into the introduction, respect to the specific goals of the paper. I would summarize the details of the whole project. Indeed, at a first reading it was little bit confusing for me, because I did not find connection in the results. Author response The referee is correct that this section is too long and distracting. We have greatly shortened this section, particularly in regard to context of the larger project. It is now nearly 25% shorter.

Referee Comments: 3. Line 136. I would specify '1 m s-1 wind condition'. Could be confusing. Author response Corrected. Thank you.

Referee Comments: 4. Line 149. I do not know how FTIR Spectrometer works in details, but based on my experience in VOCs measurements with PTR-TOF-MS and cartridges, the best option for VOCs sampling is to use PTFE (Teflon) for sampling line. This is because it is the most inert and least reactive material, avoiding the loss of the sticky compounds as many VOCs could be. It is true that the high temperatures of the gas inside the probe mitigates sticking of compounds on probe walls (showing an average gas temperature inside the probe could be useful, it is available), but I would insert few lines that would take into account consequences of using a stainless steel probe. Because of the stickiness, some compounds could be lost, they could react and become something slightly different from what is primarily emitted by pyrolysis and combustion, or they could be underestimated. This concept is valid in the same way also for the White cell. This is noticed at line 279 for the ammonia in the results. Author response This has been addressed. Stainless steel was used for the sampling probe and much of the transfer line as Teflon melts at these temperatures (> 327 C). It is also true that stainless steel is a bit more "sticky" but this is really not problematic for this application, save for amines. See the revised text.

Referee Comments: 5. Line 154-156 and Table 1. Why if in the manuscript are reported only the 21 experiments carried out in November 2018, at the beginning of the paragraph all measurement were reported? This could be confusing. I would mention only the experiments showed in the paper.

Author response The referee is correct; the misleading text that was in lines 131-133 has been removed such that only the 21 burns of interest are discussed.

Referee Comments: 6. All of Chapter 2. Experimental. Always based on my experience in VOCs sampling, I wonder how you took into account the possible contribution to VOCs identification and quantification of the Wind Tunnel, white cell and other canisters? Some compounds could be already present because released by one of these sources and not from the processes that you are surveying in this work, or both could emit them and bias your quantitative estimation. In my experience it is always needed

a blank (zero) measurement of the surveying matrix and means. This is important also for the ceramic plant holders. What they emit? What they emit when they burn? This could bias your results. Author response That is correct, but as opposed to e.g. GC, TOF-MS or other VOC sampling methods, the infrared experiment is always effectively "self-ratioing" in that the sample spectrum of intensity I is always ratioed against a zero or blank measurement Io thus having every absorbance spectrum [which is -log(I/Io)] divide out the effects of ambient background / sampling device gases. We have added a few sentences at lines 186-190 to make this more clear. The are reproduced here: "For both acquisition modes (static / dynamic), a single Io reference spectrum at the appropriate resolution was collected by flowing ambient gas into the cell at the start of each day to form the single (static) or multiple (dynamic) decadic absorbance spectra using Beer's law: - log10(I/Io). Acquiring such a blank or zero Io spectrum effectively accounts for any trace VOC emissions from the White cell, wind tunnel, tubing etc. "

Referee Comments: 7. Table 2. I would report standard deviation, since reported mixing ratios come from multiple scan averaging of 30 minutes. In this way, it is possible to observe the mixing ration variation during process observation. Author response This is a bit more difficult to address as the 30 minutes' data collection all are averaged to form just one interferogram in the FTIR software (automatically). This single interferogram is transformed to generate just one spectrum (see above). Thus, there are no multiple sets of data from which a standard deviation can be derived. As discussed in our earlier papers (Scharko et al. 2019a, b), however, what can be derived are the residuals from the MALT fit to the spectroscopic data and the multiple fits to the same spectrum using different spectral microwindows. The latter is more informative and has been shown to generate variance ranging from ca. 2 to 20% depending on the molecule and signal strength.

Referee Comments: 8. Line 466-480, paragraph 3.3 . This is a state of art about phenol emissions by burns. It should stay in introduction defining the background knowledges at the base of this study. Author response While there is some merit to this statement

that the current state of the art information about phenol could be lodged in the introduction, we respectfully disagree with this reviewer and believe that it adds more merit to the discussion to juxtapose the new results with the prior results of others. To analyze phenol was not the motivation for the experiment, it was simply one of the results that appeared upon analyzing the data and the discussion regarding these results belongs there - in the discussion.

---

## Referee Comment (RC3) · Anonymous Referee #3 · 24 Dec 2020

General comments: This paper report behavior of 29 gases emitted from biomass burning using two different FTIR acquisition modes (i.e. static and dynamic) as well as IR thermal imaging. Also, phenol temporal profile was characterized especially. I think the findings shown in this paper are suitable for AMT journal topic. I would recommend accepting this paper with a couple of revision and re-consideration mentioned below.

Specific comments: 1. Table 2: Category (d) includes not only aromatics (benzene, naphthalene, and phenol) but also furan-related compounds (furan and furfural). So, the term of this category should be "aromatics and furans".

[Figure]

2. page 15, line 355-356: Authors mentioned that they could not estimate the fractions of high-temperature and low-temperature pyrolysis from acetylene-to-furan ratio. But I think, they might be able to estimate those fractions by directly fitting the FTIR results of 29 gases using the high-temperature and low-temperature profiles (reported in Sekimoto et al. 2018). That is, (1) the fractions of 29 gases are extracted from the high-temp. and low-temp. profiles, and are normalized. (2) Then the FTIR data is linearly fitted by the normalized selected profiles (i.e. VOC_FTIR = a*High-temp + b*Low-temp). (3) Authors can know the fractions of high-temp. and low-temp. pyrolysis from coefficients "a" and "b". It should be worth to do it.

3. Figure 3: Authors obtained thermal imaging for burn progression. If they could estimate the high-temperature and low-temperature fractions according to my comment #2, how do those fractions correlate with the burn temperature derived from the thermal image?

---

## Author Comment (AC3) · 10 Jan 2021

Banach et al. Anonymous Referee #3

General comments: This paper report behavior of 29 gases emitted from biomass burning using two different FTIR acquisition modes (i.e. static and dynamic) as well as
IR thermal imaging. Also, phenol temporal profile was characterized especially. I think the findings shown in this paper are suitable for AMT journal topic. I would recommend accepting this paper with a couple of revision and re-consideration mentioned below.

Specific comments: 1. Table 2: Category (d) includes not only aromatics (benzene, naphthalene, and phenol) but also furan-related compounds (furan and furfural). So, the term of this category should be "aromatics and furans".

Author response This is a good suggestion. While many organic chemistry textbooks do in fact classify furan and related compounds as aromatics because they have a 4n + 2 aromatic system (as per Hückel's rule), furans can perhaps be considered as subgroup of related compounds as compared to e.g. benzene. We have relabeled the category should be "aromatics and furans" as suggested. Thank you.

Referee Comments: 2. page 15, line 355-356: Authors mentioned that they could not estimate the fractions of high-temperature and low-temperature pyrolysis from acetylene-to-furan ratio. But I think, they might be able to estimate those fractions by directly fitting the FTIR results of 29 gases using the high-temperature and low-temperature profiles (reported in Sekimoto et al. 2018). That is, (1) the fractions of 29 gases are extracted from the high-temp. and low-temp. profiles and are normalized. (2) Then the FTIR data is linearly fitted by the normalized selected profiles (i.e. VOC_FTIR = a*High-temp + b*Low-temp). (3) Authors can know the fractions of high-temp. and low-temp. pyrolysis from coefficients "a" and "b". It should be worth to do it. 3. Figure 3: Authors obtained thermal imaging for burn progression. If they could estimate the high-temperature and low-temperature fractions according to my comment #2, how do those fractions correlate with the burn temperature derived from the thermal image?

Author response As discussed in the manuscript, due to a) the short IR scan times and b) the weak furan infrared signals, as well as c) furan's strongest band being largely obscured by saturated $CO_2$ lines, for these data we were unable to use the metric suggested by Sekimoto et al. (2018) to assess the high- vs. low-temperature pyrolysis

as derived from the acetylene-to-furan ratio. Also, to adequately respond to suggestions #2 and #3 would represent a significant re-analysis of the entire data set as well as further interpretation and discussion. This is unfortunately not possible at this late stage: Laboratories such as ours may only work on currently funded and authorized projects. Both funding and authorization for the project ran out on 30 December. Moreover, the two scientists that performed the analysis have both left our institution for other research studies and institutions and are thus no longer available to work on the project, so our capabilities for data re-analysis at this time are limited. But as noted in the manuscript, this is part of a larger study and we plan to investigate the temperature dependent factors, possibly using the analysis methods suggested by reviewer #3, in a future work.

———————————————————

---

## Author Response (AR2)

**Response to Editor re: "Dynamic Infrared Gas Analysis from Longleaf Pine Fuelbeds Burned in a Wind Tunnel: Observation of Phenol in Pyrolysis and Combustion Phases" by Catherine A. Banach et al.**

**Received: 27 January 2021**

We thank the editor Dr. Famulari for the helpful suggestions.  We have implemented nearly all of the suggested changes.